# Unraveling the Mechanisms of Virus-Induced Symptom Development in Plants

**DOI:** 10.3390/plants12152830

**Published:** 2023-07-31

**Authors:** Tong Jiang, Tao Zhou

**Affiliations:** Department of Plant Pathology, China Agricultural University, Beijing 100193, China

**Keywords:** plant virus, virus-host interaction, cellular process, transcriptome, metabolome, chloroplast, hormone, reactive oxidative species, environmental factors

## Abstract

Plant viruses, as obligate intracellular parasites, induce significant changes in the cellular physiology of host cells to facilitate their multiplication. These alterations often lead to the development of symptoms that interfere with normal growth and development, causing USD 60 billion worth of losses per year, worldwide, in both agricultural and horticultural crops. However, existing literature often lacks a clear and concise presentation of the key information regarding the mechanisms underlying plant virus-induced symptoms. To address this, we conducted a comprehensive review to highlight the crucial interactions between plant viruses and host factors, discussing key genes that increase viral virulence and their roles in influencing cellular processes such as dysfunction of chloroplast proteins, hormone manipulation, reactive oxidative species accumulation, and cell cycle control, which are critical for symptom development. Moreover, we explore the alterations in host metabolism and gene expression that are associated with virus-induced symptoms. In addition, the influence of environmental factors on virus-induced symptom development is discussed. By integrating these various aspects, this review provides valuable insights into the complex mechanisms underlying virus-induced symptoms in plants, and emphasizes the urgency of addressing viral diseases to ensure sustainable agriculture and food production.

## 1. Introduction

Viral infections have a significant impact on plant health and can lead to a wide range of symptoms that affect growth, development, and overall productivity [1,2]. The economic impact of viral diseases on crop cultivation and food security can be substantial and far-reaching, as these diseases affect various crops in agriculture and horticulture, leading to reduced yields and lower crop quality [2,3]. Plant viruses are microscopic pathogens that infect and replicate within the cells of plants. Upon infection, plant viruses utilize various strategies to subvert the host molecular machinery and redirect resources for viral replication [4,5,6,7]. The consequences of viral infection can manifest in a variety of symptoms, including leaf discoloration, stunted growth, wilting, necrosis, and deformities in plant structures (Table 1) [2,8,9,10,11,12]. These symptoms are the visible outcomes of complex molecular interactions between the virus and its host plant.

At the molecular level, viruses have evolved sophisticated strategies to interact with and manipulate host proteins, thereby completing their life cycle within plant cells [13,14,15,16,17,18]. These interactions directly interfere with the functions of host proteins, leading to perturbations in various cellular processes. These disruptions include dysfunction of chloroplast processes, hormone manipulation, accumulation of reactive oxygen species (ROS), and alterations in cell cycle. Consequently, these molecular disturbances give rise to the emergence of viral symptoms [4,19,20,21]. Recent studies have revealed the intricate interactions between plant viruses and host factors, providing insights into the complex molecular processes underlying the development of viral-induced symptoms.

In addition to directly affecting cellular processes, viral infections can trigger a series of host responses. Upon invading a plant cell, the virus utilizes the host biosynthetic machinery to synthesize its proteins and nucleic acids, while diverting the host energy and nutrient resources to locations essential to virus replication and assembly [22,23,24]. This redirected resource allocation also disrupts the normal physiological activities of the plant cell [4,25,26,27]. At the same time, the host reprograms its metabolic pathways and gene expression patterns to adapt to the stress caused by viral infection [4,12,28]. However, this reprogramming may also have adverse effects on the physiological processes and development of the plant, thereby promoting the development of symptoms associated with viral infection [12,29].

This review also discusses the influence of environmental factors on symptoms development in plants. Light and temperature are pivotal in plant physiology and are intricately linked to various signaling pathways [30,31]. Recent studies have revealed an interplay between light or temperature and viral infection, revealing that light or temperature can modulate the severity of viral symptoms [12,32,33,34,35,36,37]. This highlights the need for a comprehensive understanding of the complex interplay among viral factors, host responses, and environmental cues.

By synthesizing these recent advances, this comprehensive review aims to provide valuable insights into the complex mechanisms underlying virus-induced symptoms in plants. Understanding these mechanisms is crucial for developing effective strategies for detecting, preventing, and controlling viral infections, ultimately safeguarding global crop production and ensuring food security in the face of emerging viral diseases.

**Table 1 plants-12-02830-t001:** Selection of 20 agronomically important plant viruses and their symptoms on crops.

Virus Name	Symptoms	Crops/Plants	References
Banana bunchy top virus	Stunted growth, twisting of leaves	Banana	[38,39]
Barley yellow dwarf virus	Yellowing, stunting, reduced grain yield	Wheat	[40,41]
Cassava mosaic virus	Mosaic, distortion, reduced root yield	Cassava	[42]
Citrus tristeza virus	Yellowing, stem pitting, leaf curling	*Citrus* spp.	[43]
Cucumber mosaic virus	Mosaic, distortion, yellowing	Cucumber	[44]
Faba bean necrotic yellows virus	Yellowing, necrosis, stunting	Faba bean	[45]
Grapevine leafroll-associated viruses	Yellowing, leaf curling, reduced fruit quality	Grapevine	[46,47,48]
Maize dwarf mosaic virus	Mosaic, stunting, yellow streaks	Maize	[49]
Maize streak virus	Yellow streaks on leaves, stunted growth	Maize	[50,51]
Papaya ringspot virus	Ring spots, leaf distortion, fruit deformities	Papaya	[52,53]
Plum pox virus	Yellow rings, fruit deformation	Stone fruits	[54,55]
Potato virus Y	Leaf mosaic, yellowing, necrosis	Potato	[56,57]
Rice tungro spherical virus	Stunting, yellowing, reduced grain yield	Rice	[58]
Rice yellow mottle virus	Yellowing, stunting, yellow mottling on leaves	Rice	[59,60]
Southern rice black-streaked dwarf virus	Stunted growth, black streaks on leaves	Rice	[61]
Sugarcane mosaic virus	Mosaic, yellow streaks, stunting	Maize	[62]
Sweet potato chlorotic stunt virus	Yellowing, stunting, leaf deformation	Sweet potato	[63,64]
Tomato spotted wilt virus	Wilted leaves, ring spots, bronzing	Tomato	[65,66]
Tomato yellow leaf curl virus	Yellowing, leaf curling, reduced fruit set	Tomato	[67,68]
Zucchini yellow mosaic virus	Mosaic, yellowing, distortion	Cucurbits	[69,70]

## 2. Alterations in Plant Cellular Processes

Emerging studies have revealed the mechanisms by which viruses manipulate plant cellular processes to induce these symptoms. Among the key factors involved, dysfunction of chloroplast proteins, hormone manipulation, ROS accumulation, and cell cycle control have been identified as critical contributors to the manifestation of virus-induced symptoms in plants (Figure 1).

### 2.1. Dysfunction of Chloroplast Proteins

Emerging studies suggest a close relationship between the appearance of symptoms and alterations in chloroplast morphology and protein function. Interactions between virus-encoded proteins and chloroplast proteins play a central role in perturbing chloroplast integrity and functionality. Viruses strategically target various chloroplast components, including photosynthetic proteins, enzymes involved in chlorophyll biosynthesis, and proteins involved in carbon assimilation and metabolism [71,72,73,74,75,76]. By impeding localization of chloroplast proteins or disrupting the function of chloroplast proteins, viral infections alter the function and structure of the chloroplast, resulting in impaired photosynthesis and altered metabolic processes. These disruptions ultimately compromise plant growth and development, contributing to the manifestation of virus-induced symptoms [77,78,79,80].

For example, in the case of Alternanthera mosaic virus (AltMV) infection in *Nicotiana benthamiana*, the excessive expression of the AltMV-encoded triple gene block 3 (TGB3) from potato virus X (PVX) leads to noticeable symptoms of veinal necrosis and chloroplast vesiculation. The interaction between AltMV-TGB3 and the Photosystem II oxygen-evolving complex protein (PsbO) is presumed to play a crucial role in symptoms development and the disruption of chloroplast structure [80,81]. One possible explanation for this observation is that the interaction between TGB3 and PsbO in the cytoplasm disrupts the normal recruitment of newly synthesized PsbO to the chloroplast and photosystem II (PS II). This disruption affects the turnover of D1, which is a crucial protein encoded by the chloroplast genome and fundamental to the assembly and functions of the PSII reaction center [82,83,84]. Consequently, the destabilization of thylakoids and PS II occurs, leading to the disruption of chloroplast structure and function.

Chlorotic and necrotic stripes are frequently observed in the leaves of rice plants infected with rice stripe virus (RSV), leading to subsequent wilting of the affected plants. Research findings indicate that expression of RSV-encoded disease-specific protein (SP) in transgenic rice enhanced RSV induced symptoms. This intensified effect is attributed to the interaction between SP and a component of the luminal protein complex associated with photosystem II (PsbP). In the presence of RSV SP, a substantial portion of PsbP is recruited into the cytoplasm. The absence of PsbP, as one of the key host factors involved in chloroplast function in the chloroplast, can lead to degradation of photosynthetic pigments, including chlorophyll [85,86]. Moreover, the accumulation of SP during RSV infection led to notable alterations in both the structure and functionality of chloroplast. These changes collectively contribute to the severity of RSV-induced symptoms [73].

Cucumber mosaic virus (CMV) infection leads to the manifestation of mosaic symptoms on various host plants. The coat protein (CP) serves as the determinant of CMV-M induced symptoms [75]. Subsequent analysis by other groups revealed that CMV-M-encoded CP interacts with the precursor of chloroplast ferredoxin I (Fd I) in the cytoplasm and disrupts the transport of Fd I into chloroplast. This altered localization of Fd I impacts its function in multiple ways. First, it disrupts electron transport in the plant host, leading to the accumulation of H_2_O_2_ in the chloroplast [87,88,89]. The substantial oxidative stress caused by H_2_O_2_ might be toxic to the chloroplast [90,91]. Second, the reduced presence of Fd I in the chloroplast affects various metabolic activities, including the biosynthesis of chlorophyll and phytochrome [87]. As a result, the disrupted function of Fd I during CMV-M infection ultimately results in the development of chlorosis symptoms on tobacco leaves [75,91].

Apart from perturbing the subcellular localization of chloroplast proteins, CMV can also interfere with chloroplast genes to induce specific symptoms. The Y satellite RNA (Y-Sat) of the CMV produces short interfering RNAs (siRNAs) that target the mRNA of chlorophyll biosynthesis genes (*CHLI*). Downregulation of *CHLI* mRNA by Y-Sat-derived siRNAs induces yellowing symptoms in infected plants [8,92].

Chlorosis symptoms induced by potato virus Y (PVY) infection on host plants are attributed to disruptions in chloroplast structure and function caused by interaction between the viral protein and host proteins. Specifically, the PVY-encoded HC-Pro interacts with the chloroplast division-related factor MinD. This interaction inhibits dimerization process of the MinD protein, consequently perturbing its functional activity and interfering with chloroplast division. This disruption leads to changes in chloroplast number within the host cells, contributing to symptoms development [93].

These research findings provide compelling evidence that the function of chloroplast proteins plays a critical role in the development of virus-induced symptoms in plants.

### 2.2. Hormone Manipulation

Hormones serve as chemical messengers, orchestrating diverse cellular responses throughout a plant lifecycle [94,95,96]. By hijacking and manipulating key components of hormone signaling, viral proteins can disrupt the delicate balance of hormonal cues, tipping the scales towards aberrant growth patterns and developmental abnormalities [97,98,99,100].

Dwarf symptoms are frequently observed in rice plants infected with rice dwarf virus (RDV). The interaction between RDV-encoded P2 protein and rice *ent*-Kaurene oxidase leads to a reduction in the activity or transcription of *ent*-Kaurene oxidase. This, in turn, results in a decrease in endogenous gibberellin (GA) content within RDV-infected rice plants. The reduced accumulation of GAs directly contributes to the abnormal function of GA-regulated cellular processes, which ultimately leads to dwarf symptoms expression [101]. Additionally, P2 interacts with the OsIAA10, a rice auxin/indole-3-acetic acid protein, to prevent the interaction between OsIAA10 and transport inhibitor response1 (OsTIR1), leading to the inhibition of OsIAA10 degradation by the 26S proteasome. Consequently, this disruption results in a reduced sensitivity of RDV-infected rice plants to auxin signaling. The reprogramed auxin signaling causes disease symptoms including dwarfing, increased tiller number, and short crown roots in infected rice [98].

In *Arabidopsis thaliana* ecotype Shahdara, infection with tobacco mosaic virus (TMV) induces a distinct set of disease symptoms, such as stunting, necrosis of the inoculated leaf, loss of apical dominance, and leaf curling. The 126/183 kDa replicase of TMV was found to interact with PAP1, a putative regulator of auxin response genes. This interaction disrupts the nuclear localization of PAP1, thereby affecting its transcriptional regulatory function on genes involved in the auxin response pathway. As a consequence, the gene expression patterns associated with the auxin response system are perturbed, leading to the manifestation of specific disease symptoms in infected plants [97].

*N. benthamiana* plants infected with ageratum leaf curl Sichuan virus (ALCScV) showed dwarfing symptoms and abnormal flower development. ALCScV-encoded C4 interacts with DELLA proteins (NbGAI) that act as inhibitors of plant growth and development, particularly in the GA signaling pathway, where they negatively regulate GA biosynthesis [102,103]. The interaction between C4 and NbGAI disrupts the normal interaction between NbGAI and another protein called gibberellin insensitive dwarf 2 (NbGID2). This disruption prevents the degradation of NbGAI, ultimately inhibits the GA signaling pathway, leading to dwarfing symptoms and abnormal flower development in the infected plants [99].

The disruption of hormone signaling pathways by viral proteins provides insights into the mechanisms underlying the development of disease symptoms in infected plants. Understanding these interactions can contribute to the development of strategies for disease management and the improvement of crop resilience.

### 2.3. ROS Accumulation

ROS exhibits dual functionality in the invasion of pathogens in plant cells. Low levels of ROS have been demonstrated to promote signal transduction and acclimation/defense responses, whereas high levels of ROS lead to oxidation damage to lipids, DNA, and proteins [104,105,106,107,108]. This oxidative damage can disrupt cellular functions and impair cellular components, ultimately resulting in cellular death and the emergence of necrosis or mosaic symptoms observed during viral infections.

For example, the appearance of necrotic spots in *Arabidopsis* plants infected with CMV is correlated with an elevation in H_2_O_2_ production. This effect is attributed to the interaction between the CMV 2b and the isoenzyme catalase 3 (CAT3) within infected tissues, which results in the inhibition of catalase activity. This inhibition subsequently triggers oxidative stress, leading to the specific induction of necrosis [109].

Maize chlorotic mottle virus (MCMV) causes chlorotic and mottle symptoms, and even necrosis when it infects maize [110]. The MCMV-encoded protein P31 is essential for necrosis symptoms development by targeting and inhibiting the activity of catalase. Inhibition of catalase activity by P31 results in increased accumulation of H_2_O_2_ with infected tissues, consequently inducing necrosis [62].

Bamboo mosaic virus (BaMV) causes chlorotic mosaic symptoms in both *Brachypodium distachyon* and *N. benthamiana*. Studies have revealed that BaMV infection resulted in a notable accumulation of full-length Cu/Zu superoxide dismutase preprotein (PrNbCSD2) within the cytosol, which may be achieved by impairing the cleavage of PrNbCSD2 transport peptide. This accumulation of PrNbCSD2 protein led to an excessive production of H_2_O_2_ in BaMV-infected cells, consequently resulting in the manifestation of chlorotic symptoms [111].

Mosaic symptoms caused by sugarcane mosaic virus (SCMV) infection were found to correlate with ROS accumulation in mitochondria. SCMV enhances the enzymatic activity of pyruvate orthophosphate dikinase (PPDK) at the infection front, leading to excessive production of malate under light, which in turn activates the malate circulation pathway and causes the ROS accumulation in mitochondria, thereby causing the manifestation of mosaic symptoms [12].

Overall, these studies collectively reveal the mechanisms through which viral infections induce ROS imbalance, underscoring the significance of ROS imbalance in the pathogenic processes of viral infections.

### 2.4. Cell Cycle Control

Cell division serves as the foundation for plant growth, enabling the formation of new tissues and organs throughout the plant life cycle [112]. However, DNA viruses-encoded proteins can disrupt the intricate machinery that regulates cell division, creating an environment conducive to viral replication. This disruption can result in abnormal cell growth, uncontrolled proliferation, and ultimately, tissue deformities in infected plants [113,114,115].

Members of the plant retinoblastoma-related proteins (RBR) family serve as negative regulators of the cell cycle [116]. Several DNA viruses-encoded proteins, such as tomato golden mosaic virus (TGMV) Rep proteins, Faba bean necrotic yellow virus (FBNYV) Clink protein, and maize stripe virus (MSV) RepA protein, have been identified to interact with RBR proteins. These interactions have significant implications for cell cycle control. TGMV Rep proteins inhibit the activity of pRBR through their interaction, leading to the entry of infected cells into the S phase where DNA synthesis occurs. FBNYV Clink protein promotes the degradation of pRBR through the proteasome, thereby exerting control over the cell cycle. Wheat dwarf virus (WDV) RepA protein stimulates cell division and calli growth of maize culture by impairing the cell cycle arrest of RBR [113,114,117].

In the case of beet curly top virus (BSCTV), the viral protein C4 has been identified as the symptom determinant. C4 induces the accumulation of RKP protein, which subsequently interacts with the cell-cycle inhibitor ICK/KRP proteins. This interaction leads to alterations in the cell cycle of infected plants. However, the precise mechanism by which C4 promotes the accumulation of RKP is not yet fully understood and requires further investigation [118].

Tomato leaf curl Yunnan virus (TLCYnV) is known to induce severe developmental abnormalities in plants. Research has shown that the C4 encoded by TLCYnV functions as one of viral symptoms determinants. This protein interacts with glycogen synthase kinase 3 (NbSKη), leading to the relocalization of NbSKη from the nucleus to the membrane and subsequently reducing its nuclear accumulation. This disruption impairs phosphorylation-dependent degradation of NbCycD1;1 (a protein that plays a crucial role in regulating regulate the G1/S-phase transition of cell cycle) [119,120,121], thereby leading to the accumulation of NbCycD1;1 in TLCYnV infected plants, inducing abnormal cell division in plants [115]. The manipulation of the cell cycle by these DNA viruses leads to excessive cell division and abnormal tissue growth, ultimately resulting in the observed tissue deformities in infected plants.

In this section, we explored how viruses manipulate plant cellular processes to induce symptoms, focusing on dysfunction of chloroplast proteins, hormone manipulation, ROS accumulation, and cell cycle control. These alterations play crucial roles in the manifestation of virus-induced symptoms in plants, providing valuable insights into the pathogenic mechanisms of viral infections.

## 3. Host Metabolic and Genetic Responses to Virus-Induced Symptoms

Plant viruses induce disturbances in host metabolic processes and gene expression, manifesting as visible symptoms that hinder the growth and development of infected host plants [122,123,124,125,126]. This section explores the host responses to metabolic and genetic alterations that accompany the appearance of virus-induced symptoms.

### 3.1. Metabolic Disorders

One classic study investigated metabolic changes in zucchini plants infected with CMV. Spatial analysis revealed distinct metabolic alterations both at the edges of advancing lesion and inside the lesion. At the edges of the advancing lesion, there was a high demand for viral replication and synthesis of viral proteins, resulting in increased activity of anaplerotic enzymes, enhanced photosynthesis, and starch accumulation [29]. The reasons behind CMV causing these metabolic disruptions can potentially be understood by studying other virus-host interactions. An increase in the activity of anaplerotic enzymes is believed to play a crucial role in replenishing intermediates of the tricarboxylic acid (TCA) cycle, providing the necessary building blocks for viral protein synthesis and energy production [127,128,129]. The enhanced photosynthesis is a result of the virus exploiting host photosynthetic machinery to generate energy and metabolic precursors required for viral replication [71,74]. Additionally, the accumulation of starch serves as an energy reservoir to support the energy demands of viral replication and protein synthesis [130]. This phenomenon suggests that CMV strategically harnesses the host resources to support its replication and protein synthesis at the edges of advancing lesion.

Inside the lesions, there was a decrease in viral proliferation, accompanied by reduced photosynthesis and metabolism of starch. As the virus reaches its peak or starts to decline in this region, the demand for host resources decreases, leading to a reduction in photosynthetic activity [29]. The decreased metabolism of starch may result from the altered cellular processes and resource allocation in response to the viral infection [29]. Concurrently, the host metabolism shifted towards glycolysis and mitochondrial respiration [29]. This metabolic reprogramming can be attributed to the virus obtaining the energy requirements in symptomatic tissue to sustain its replication [23]. Glycolysis provides a rapid energy supply through the breakdown of glucose, while mitochondrial respiration produces ATP through oxidative phosphorylation [131]. This shift in energy metabolism allows the host plant to adapt to the changing metabolic demands induced by the viral infection. These changes spatially coincide with the emergence of chlorosis symptoms [29].

A study was conducted to investigate the processes underlying the symptom development in the interaction between potato plants and PVY by examining metabolite levels [132]. In PVY-infected potato leaves, the levels of carbohydrates and substances that scavenge ROS initially decreased at 1 day post-infection (dpi), but later increased at 6 dpi. Similar dynamics were also observed for several amino acids, intermediates of the GABA shunt and TCA cycle, and phenylpropanoids [132]. In the early stages after PVY infection, the virus actively modulates the host cellular processes to facilitate its replication and spread. This manipulation includes redirecting host resources and altering metabolic pathways to meet the demands of viral replication [1,133,134]. As a result, the levels of carbohydrates and ROS scavenging substances initially decrease at 1 dpi, indicating that the virus is redirecting these resources for its own replication [132]. However, as the infection progresses and viral multiplication occurs, the plant defense responses are triggered [19,135]. ROS are generated as a part of the plant immune system to combat pathogens [104]. The subsequent increase in ROS scavenging substances at 6 dpi reflects the plant’s attempt to counteract the oxidative stress induced by the virus [132]. The accumulation of carbohydrates and other metabolites, including amino acids, intermediates of the GABA shunt and TCA cycle, and phenylpropanoids, coincides with the expected onset of virus multiplication. These metabolic changes are part of the plant defense response and recovery processes, aimed at mitigating the damage caused by the virus and restoring normal cellular function [4,132,136].

Another study focused on SCMV infection and its impact on respiratory metabolism pathways. Targeted metabolomic analysis revealed a significant induction of metabolites within respiratory metabolism pathways during the development of mosaic symptoms. Notably, there was a remarkable accumulation of malate in response to SCMV infection. Malate over-production provides energy resources for viral infection and serves as a biochemical basis for symptoms manifestation [12].

These studies collectively demonstrate the complex interplay between viral infection and host metabolism. The metabolic changes observed in virus-infected plants are attributed to the manipulation of host metabolism by virus and the subsequent response of the host plant defense mechanisms. Simultaneously, these alterations in metabolism contribute to the development of symptoms in infected plants.

### 3.2. Gene Expression Changes

The development of viral symptoms results from complex molecular and physiological processes. Thus, transcriptomic analysis of various viral infections in different plant species has provided valuable insights into the changes in gene expression associated with symptom development. 

Recent studies have investigated changes in host gene expression after the appearance of viral symptoms. In *Arabidopsis* infected with plum pox virus (PPV), genes related to soluble sugar, starch and amino acid, intracellular membrane/membrane-bound organelles, chloroplast, and protein fate were upregulated, while genes associated with development/storage proteins, protein synthesis and translation, and cell wall-associated components were downregulated. These changes in gene expression are closely linked to the infection process and the subsequent development of symptoms [137]. Tomato plants infected with tomato leaf curl New Delhi virus (ToLCNDV) exhibited changes in gene expression related to increased respiration rates, decreased photosynthesis, accumulation of soluble sugars/starch, and elevated amino acid synthesis during symptoms development [138]. Transcriptomic analysis in tomato plants infected with tomato chlorosis virus (ToCV), tomato yellow leaf curl virus (TYLCV), and co-infection of ToCV and TYLCV revealed common activation of plant-pathogen interaction and metabolic pathways across all three infection conditions. However, various pathways associated with photosynthesis, crucial for energy production and plant growth, were commonly inhibited in all three infections. This inhibition likely contributes to the chlorosis and yellowing symptoms induced by ToCV and TYLCV, respectively. The analysis further showed specific enrichment of up-regulated differentially expressed genes (DEGs) in ToCV-infected leaves related to plant pigment metabolism, such as flavonoid and anthocyanin biosynthesis. In TYLCV infection, up-regulated pathways were linked to processes including plant hormone signal transduction, regulation of autophagy, and endocytosis. These pathways may contribute to the observed symptoms induced by TYLCV or the combined infection [125].

Studies have also explored changes in host gene expression at different stages of symptom development. Transcriptome analysis comparing healthy and infected plants at six stages of symptom development (vein clearing, mosaic, severe chlorosis, partial recovery, complete recovery, and secondary mosaic) in *Nicotiana tabacum* infected with CMV revealed differential gene expression profiles. In the early stages of infection, CMV suppresses the expression of energy metabolism and pigment metabolism in the host plants. Several up-regulated pathways were related to the metabolism of terpenoids and polyketides, which might regulate plant defense [139]. During the recovery phase, DEGs were significantly enriched in pathways related to monoterpenoid biosynthesis, flavonoid biosynthesis, cysteine and methionine metabolism, and plant-pathogen interactions, indicating their potential roles in symptoms recovery [84,140]. In the secondary pathogenesis process, DEGs were enriched in pathways related to metabolism and genetic information processing. Some pathways overlapped with those in the vein clearing stage, whereas others were similar to those in the complete recovery stage, involving multiple molecular processes. This suggests that the secondary infection process is complex, resembling both the initial disease onset process immediately after viral infection and the symptoms recovery process. Overall, the results indicate a suppression of photosynthesis and pigment metabolism in both the initial and secondary phases of pathogenesis, while the transient recovery period exhibits a significant enhancement of the innate immunity process [140].

In maize infected with SCMV, transcriptome analysis was conducted to analyze the differential gene expression in both the pre-symptomatic infection stage and the steady-symptom infection stage. In pre-symptomatic infection, the function of DEGs is mainly related to protein processing in the endoplasmic reticulum, for fatty acid biosynthesis, and for RNA splicing. During steady-symptom infection stage, the function of DEGs is mainly related to proteins of the thylakoid and other plastid-related cellular components [126]. Through the analysis of changes in gene expression during the critical period of symptoms development, it was found that SCMV upregulated the malate circulation pathway associated with respiratory metabolism and downregulated genes involved in photosynthetic metabolism during symptom manifestation [12].

Based on the similarity of transcriptional changes in different viruses during the induction of host symptoms development, it is suggested that the increased energy demand during viral infection has a significant impact on energy metabolism, respiration, and photosynthetic rates. The increased energy demand during viral infection has a significant impact on energy metabolism, respiration, and photosynthetic rates. This leads to the accumulation of ROS, which can directly induce viral symptoms and act as stress signals to initiate defense pathways. Moreover, the disruption of phytohormonal balances due to shared intermediates between stress signaling and hormone synthesis greatly affects the physiology of the host, possibly related to photosynthetic rates and chloroplast reactions. The cumulative effects of inhibited chloroplast functions may be associated with stunted growth and poor development of leaves (Figure 2).

The host metabolic and genetic responses to virus-induced symptoms are complex processes that involve alterations in metabolic pathways and changes in gene expression. Viral infections strategically manipulate host metabolism to support their replication, leading to changes in carbohydrate utilization, ROS accumulation, and energy production. These alterations, along with shifts in gene expression, contribute to the manifestation of visible symptoms in infected plants. Understanding these host responses provides valuable insights into the pathogenesis of viral infections and can aid in the development of effective disease management strategies.

## 4. The Relationship between Environmental Factors and Virus-Induced Symptoms

Light and temperature are important environmental factors that profoundly influence plant growth, development and physiology. Interestingly, many studies have described the influence of light and temperature on development of viral symptoms. This section summarizes the effects of light and temperature conditions on the development of viral symptoms (Figure 3).

### 4.1. The Role of Light Conditions in Virus-Induced Symptom Development

Studies investigating the effect of light conditions on viral symptoms development have revealed intriguing findings. In the case of potato mop-top virus (PMTV) infection in tobacco plants, it was observed that the appearance of necrotic spots or small rings was delayed when there was a decrease in light intensity or photoperiod. When inoculated plants are transferred from light to darkness, necrotic rings develop, and the rate of virus accumulation increases. However, when the order of the treatments is reversed, no lesions appear. Therefore, the process of PMTV-induced lesion formation includes an early phase requiring light [33].

Similarly, a field survey conducted in an unmanaged forest revealed interesting observations regarding viral symptoms in shade-grown and sun-exposed plants. Virus-infected plant species growing under shade exhibited fewer apparent viral symptoms, while those growing in direct sunlight displayed severe chlorosis in their leaves [32].

To understand these observations, it is important to consider the role of light in photosynthesis. When a plant is infected with a virus, it can affect the photosynthetic machinery, leading to changes in energy production [71,74]. Alterations in photosynthesis can disrupt various physiological processes in plants, ultimately influencing the occurrence of symptoms. Recent research on maize plants infected with SCMV further emphasized the role of light in viral symptoms development. It was found that mosaic symptoms caused by SCMV infection only occurred under light illumination, and no mosaic symptoms were observed under dark or low-light conditions. Additionally, it was discovered that SCMV infection induced the overproduction of malate that could cause symptoms via elevating PPDK enzyme activity under light [12]. In maize plants, the key regulatory factor governing the response of PPDK to light is PPDK Regulatory Protein (PDRP) [141,142]. In the presence of light, PDRP catalyzes the dephosphorylation of PPDK, leading to the activation of its enzymatic activity, and consequently, promoting the conversion of phosphoenolpyruvate to oxaloacetate, and ultimately, to malate [141,142,143]. Conversely, under dark conditions, PDRP mediates the dephosphorylation of PPDK, resulting in a reduction of its enzymatic activity [141,142]. However, when maize plants are infected with SCMV, there is a potential that the virus might interfere with the regulatory function of PDRP. By affecting the phosphorylation state of PPDK, SCMV could disrupt the normal regulation of PPDK under light conditions, leading to abnormal levels of malate and PPDK enzyme activity. Thus, investigating whether SCMV can modulate PPDK phosphorylation and enzymatic activity via its impact on PDRP functionality represents an intriguing avenue of research. Overall, this finding provides a partial explanation for how light affects the development of viral symptoms [12].

The above research indicates the intricate relationship between light conditions and the development of symptoms. It is worth noting that the specific effects of light on plant virus symptoms can vary depending on the virus-host combination, light intensity, duration, and other environmental factors. The specific mechanism by which light affects the development of virus symptoms is still being investigated, but these findings highlight the importance of considering light conditions when studying plant-virus interactions and symptoms development.

### 4.2. The Role of Temperature in Virus-Induced Symptoms Development

Temperature is a crucial factor in determining the outcome of plant virus infections. However, the effects of increased temperatures on viral symptoms can vary depending on the specific virus-host combinations. Some studies have shown that higher temperatures can intensify virus symptoms [144], while in other cases, a phenomenon called “heat masking” occurs, where symptoms are reduced or eliminated despite the host remaining infected [145]. Recent research has described both positive [36,146] and negative [34,35,37,147] correlations between temperature and the severity of plant viral diseases. However, the underlying molecular mechanisms behind these phenomena are still not fully understood.

For example, in *N. tabacum* plants infected with CMV, higher incubation temperatures (28 °C compared to 18 °C) resulted in more severe symptoms. Molecular analysis of CMV-infected plants revealed that at lower temperatures, several genes associated with salicylic acid (SA) were upregulated, while at higher temperatures, genes associated with jasmonic acid (JA) showed increased expression [148]. SA-dependent responses are typically associated with plant defense against biotrophic pathogens, whereas JA pathways are often activated in response to necrotrophic pathogens. It is important to note that these two pathways, SA and JA, are antagonistic to each other. This means that when JA pathways are induced, plants become more susceptible to biotrophic pathogens, whereas the induction of SA pathways makes plants more vulnerable to necrotrophic pathogens.

At 25 °C, a severe strain of AltMV causes mosaic patterns and localized necrosis in *N. benthamiana* plants. However, when the temperature drops to 15 °C, these symptoms progress to systemic necrosis, resulting in plant death within 30 dpi. By contrast, a chimera consisting of the CP derived from a mild AltMV strain only induces systemic mosaic symptoms at 15 °C, reducing the severity of the symptoms caused by the severe strain. Surprisingly, there was no significant difference in virus accumulation between the severe and mild strains at a given temperature. However, it was observed that both strains exhibited significantly higher virus accumulation at 15 °C when compared to 25 °C [149]. This observation aligns with the trend of symptoms severity being correlated with the absolute titer of the symptoms-inducing factor. Further investigation revealed that the CP form responsible for inducing systemic necrosis interacts with a boron transporter. This interaction likely disrupts boron metabolism within the plant, ultimately triggering systemic necrosis.

These studies indicate that light and temperature can have diverse effects on plant viral infections and the severity of associated symptoms. The interplay between environmental factors, plant defense pathways, and viral factors contributes to the complex molecular mechanisms underlying these phenomena. Nevertheless, it is essential to note that beyond light and temperature, other environmental factors, such as CO_2_ concentration, UV radiation, ozone, and drought, can also influence the interactions between plant and virus [150,151,152,153]. Despite the progress made in this field, there is still a lack of comprehensive and in-depth studies, and a comprehensive understanding of these mechanisms and their implications for plant-virus interactions remains to be fully elucidated. Researchers need to conduct more comprehensive studies to gain a deeper insight into the impact of environmental factors on plant viral infections and how they affect the severity of symptoms.

## 5. Conclusions

Understanding the complex mechanisms by which viruses interact with host factors and manipulate cellular processes is essential for unraveling the causes of virus-induced symptoms in plants. Recent research has provided valuable insights into these intricate processes, revealing the mechanisms by which viral infections disrupt chloroplast protein function, hormone regulation, ROS accumulation, and cell cycle control. Additionally, host reactions such as metabolic disorders and gene expression changes contribute to symptoms development. The influence of environmental factors on viral infection and symptoms severity further emphasizes the multifaceted nature of virus-plant interactions. By integrating these findings, this comprehensive review offers a holistic understanding of the mechanisms underlying virus-induced symptoms in plants, paving the way for the development of effective strategies for disease management and crop protection strategies.

By gaining a deeper understanding of how viruses disrupt plant cellular processes, novel intervention strategies can be explored to mitigate the impact of viral infections on crops. Building upon our understanding of the metabolic disorders and gene expression changes induced by viral infections, targeted approaches can be developed to enhance the crops resistance to diseases. One of the cutting-edge technologies that has emerged in recent years is the use of RNA-based approaches, such as dsRNA and siRNA gene interference, microRNA, and CRISPR/Cas9. These RNA-based techniques offer promising methods for developing virus-resistant crop plants. By introducing specific RNA molecules that target viral genes or disrupt viral replication, these approaches can effectively inhibit viral infections in plants. Moreover, the development of non-transgenic methods, such as topical application of dsRNA, hairpin RNA, and artificial microRNA, provides GMO-free virus disease management options, which are increasingly sought after by consumers and regulatory authorities [154,155]. Another promising technology is the use of gene-editing tools like CRISPR/Cas9. This powerful tool allows precise modifications to plant genomes, enabling the introduction of resistance genes or the disruption of viral genes. CRISPR/Cas9-based strategies offer a targeted and efficient approach to engineer virus-resistant crops without introducing foreign DNA, thus addressing concerns related to the use of genetically modified organisms [154,155]. Furthermore, advancements in high-throughput sequencing and bioinformatics have enabled more rapid and accurate identification of viral strains and variants. This information is invaluable for implementing timely and effective disease management strategies, as it allows for the monitoring of viral diversity and the development of region-specific approaches to combat viral outbreaks [156]. Additionally, exploring the interaction between virus infection and environmental factors will be an important direction for future research. Investigating how environmental factors modulate plant responses to viral infections and developing methods to utilize environmental factors for regulating crop disease resistance will open up new avenues for designing more intelligent virus management strategies.

In conclusion, through in-depth research on virus-plant interactions, we will gain a better understanding of the mechanisms underlying virus-induced symptoms in plants and provide more effective strategies for disease management and crop protection. This will have significant implications for the sustainable development of agriculture and food security.

## Figures and Tables

**Figure 1 plants-12-02830-f001:**
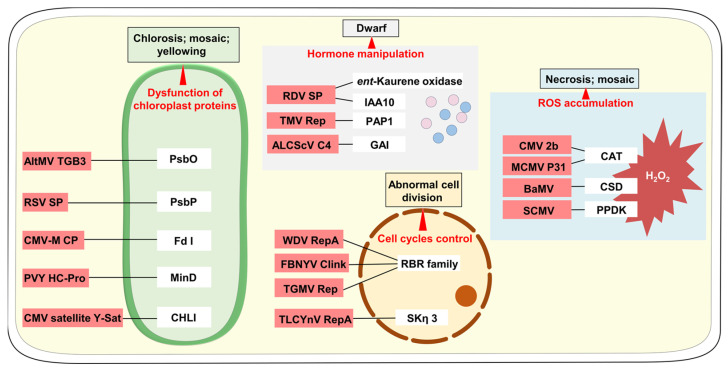
Key discoveries on the impact of plant viruses on cellular processes facilitating symptoms development through interactions with host proteins. Plant viruses disrupt various cellular functions by interacting with host proteins, including dysfunction of chloroplast proteins, hormone manipulation, accumulation of reactive oxygen species (ROS), and cell cycle control, which contribute to the development of different types of symptoms. PSbO, Photosystem II oxygen-evolving complex protein; PSbP, luminal protein complex associated with photosystem II; Fd I, chloroplast ferredoxin I; MinD, chloroplast division-related factor; CHLI, chlorophyll biosynthesis gene; IAA10, a rice auxin/indole-3-acetic acid protein; PAP1, a putative regulator of auxin response genes; GAI: DELLA proteins; RBR: retinoblastoma-related proteins; SKη: glycogen synthase kinase; glycogen synthase kinase 3; CAT: catalase; CSD: Cu/Zu superoxide dismutase; PPDK: pyruvate orthophosphate dikinase; TGB3: triple gene block 3; SP: disease-specific protein; CP: coat protein; AltMV, Alternanthera mosaic virus; RSV, rice stripe virus; CMV, cucumber mosaic virus; PVY, potato virus Y; RDV, rice dwarf virus; TMV, tobacco mosaic virus; ALCScV, ageratum leaf curl Sichuan virus; WDV, wheat dwarf virus; FBNYV, faba bean necrotic yellow virus; TGMV, tomato golden mosaic virus; TLCYnV, tomato leaf curl Yunnan virus; MCMV, maize chlorotic mottle virus; BaMV, bamboo mosaic virus; SCMV, sugarcane mosaic virus.

**Figure 2 plants-12-02830-f002:**
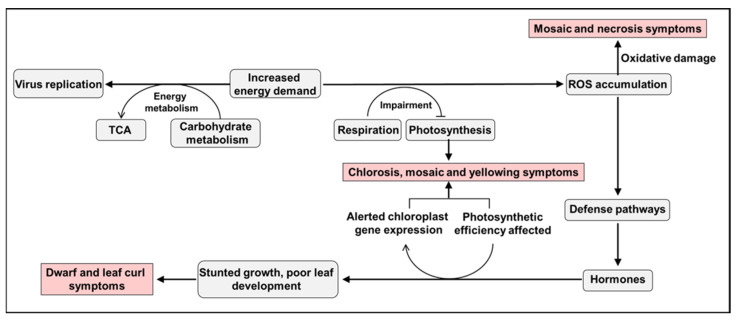
The possible models of interactions among different cellular molecular functional pathways result in the manifestation of symptoms during viral infection. During virus infection in plant cells, the replication of the virus increases the energy demand, thus impacting energy metabolism. This leads to an upregulation of host respiratory metabolism and a downregulation of photosynthesis, resulting in elevated levels of reactive oxygen species (ROS). The high levels of ROS, as highly reactive molecules, can cause damage to cellular components, leading to symptoms such as chlorosis or necrosis. Furthermore, ROS can act as signaling molecules to regulate the synthesis and signal transduction of plant hormones. Altered hormone synthesis can hinder host development and photosynthesis, resulting in symptoms such as dwarfing and leaf curling. The impairment of chloroplast function and gene expression during viral infection contributes to symptoms such as chlorosis, mosaic, and yellowing.

**Figure 3 plants-12-02830-f003:**
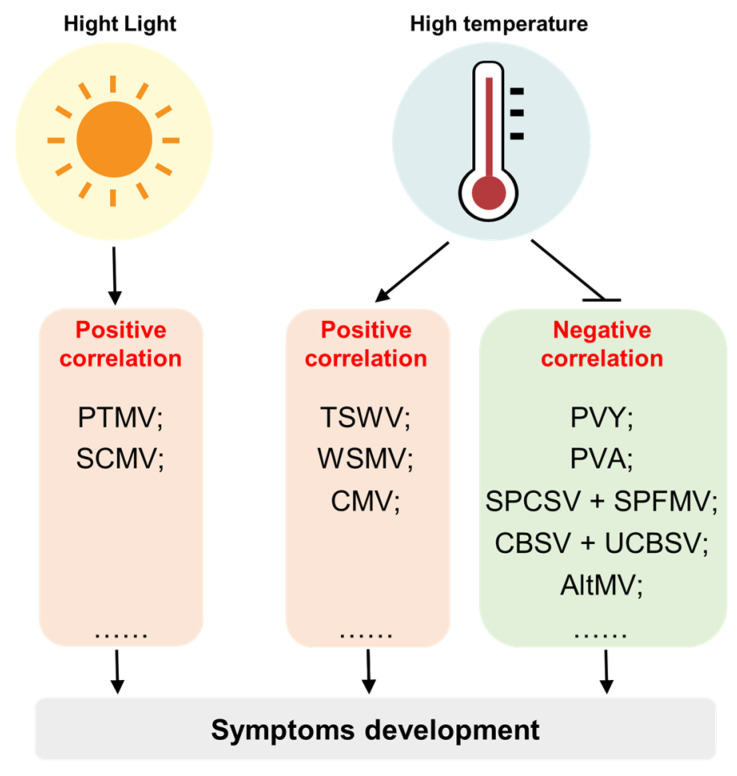
Environmental factors that influence the development of viral symptoms include light and temperature. In general, there is a positive correlation between light intensity and the severity of viral symptoms. However, the correlation between temperature and viral symptoms can vary depending on the specific virus and host. For some viruses like tomato spotted wilt virus (TSWV), wheat streak mosaic virus (WSMV), and cucumber mosaic virus (CMV), higher temperatures are associated with enhanced symptoms development, indicating a positive correlation. On the other hand, certain viruses like potato virus Y (PVY), potato virus A (PVA), sweet potato chlorotic stunt virus + sweet potato feathery mottle virus (SPCSV + SPFMV), cassava brown streak virus + Ugandan cassava brown streak virus (CBSV + UCBSV), and alternanthera mosaic virus (AltMV) induced symptoms show a negative correlation with temperatures, where high temperatures inhibit the symptoms development.

## Data Availability

No new data were created or analyzed in this study. Data sharing is not applicable to this article.

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
