# Peer review of "Unraveling the Mechanisms of Virus-Induced Symptom Development in Plants"

_plants, 2023, doi:10.3390/plants12152830_

Round 1

Reviewer 1 Report

-        Line 86: add explanation of PVX

-        The manuscript needs to be reorganized.

-        Several sentences and paragraphs missed reference citations.

-        Many sentences are unclear and need to be clarified.

-        Line 440: What means by above researchers in " Above researches indicate the intricate relationship between light conditions................"

-        Line 453: What means by "Modern research"; a modern word, not an appropriate word

-        References need to update

Many sentences are unclear and need to be clarified.

Reviewer 2 Report

Plant viruses remain a significant impediment in successful cultivation of plants and interfere with growth and development. Elucidation of plant-virus interactions is important to gain understanding of disease development process and formulate strategies for viral manage viral infections.

In this review MS authors have highlighted the various plant-virus interactions and provided its impact on plant physiology.

Overall, the review is timely and comprehensive report on plant viruses.

Some comments to improve the MS:

1.       Authors should provide some statistical information on viral diseases in abstract.

2.       Additionally authors need to consider economic impact of viral disease on crop cultivation and food security.

3.       Key genes that increase virulence needs to be mentioned and discussed in abstract.

4.       Authors should discuss some latest techonology for virus managelment in plants, for example, PMID: 34975291

5.       Adding a table with list of viruses, symptoms, crops/plants would be good addition for this review.

Okay.

Reviewer 3 Report

Title: Understanding How Viruses Cause Symptoms in Plants

This is a review paper that explores the mechanisms behind the development of virus-induced symptoms in plants. The authors have organized the paper into three main sections which are: the changes in plant cellular processes, the metabolic and genetic responses of the host to virus-induced symptoms, and the relationship between environmental factors and virus-induced symptoms. Additionally, they have included three figures that summarize the fundamental discoveries of virus impact on cellular processes, the proposed interaction models, and the correlations between symptom development and environmental factors. The paper cites 104 references and discusses at least 21 viruses that affect herbaceous plants or crops. However, the authors did not mention any viruses affecting woody plants such as Citrus or Grapevine. This was likely a deliberate choice to focus on viral symptoms and disease development stages that have been more extensively studied using Nicotiana benthamiana as a model species.

Although sections 2 and 4 reflect a critical evaluation and an in-depth analysis of the literature content, in my view section 3, regarding host metabolic and genetic responses, is less accomplished. As a review paper, a more systematic approach to the literature on this topic is expected. It is true that the information on host responses can still be limited, at least for some viruses, and mostly host-specific. So, it would be advantageous to revise this section to be less descriptive and more focused on organizing and systematizing the literature findings. Particularly, it would be crucial to distinguish between gene expression changes that are transversal and those that are host specific.

I would appreciate it if the authors could address the following comments, suggestions, and queries and provide a response. 

Line 46

Lines 49-50

Lines 57-58

Line 84 – Alternanthera mosaic virus

Line 540 – is this reference complete?

Line 93 – “essential for the function of PSII” sounds too bland…D1 protein is fundamental to the assembly and functions of the PSII reaction center, so maybe a more informative sentence could be included here.

Line 107 – “Qiu” needs the reference: for CMV Qiu et al. 2018 is reference number 41 – is this correct?

Line 108 –“Subsequent analysis revealed that…” – the text here is including data from other references. Please change to: “Subsequent analysis by other authors revealed that…”.

Line 203 – “Cell cycles control “ – change to ”Cell cycle control”

Lines 261-285 – this part is describing the spatial analysis of metabolic alterations caused by CMV in zucchini plants. However, data from other viruses (like TBSV) and other hosts are inserted in this discussion without the corresponding references by its side, making it confusing. Could this be better articulated, without losing the information?

Lines 286-304 – evidence discussed here is all from reference 95. The information given is disconnected from the previous text. 

Lines 332-354 – only describe reference 89. Could this be better linked with the previous and subsequent paragraphs?

Lines 361-362: 364-365– there is no reference to accompany the statements given in these lines.

Lines 366-367- The sentence is confusing.

Line 375 – Please correct the repetition.

Lines 377 and 379 – what are “differential genes”? Do the authors must mean “differentially expressed genes”? Then please use clear naming.

Line 380 – is reference 90 the correct one here?

Line 386-387- please delete the part “we have provided a genome-wide transcript profile, and their functional annotation… in virus-infected plants. “. The authors did not do this! 

Suggested text: “Based on the similarity of transcriptional changes caused by different viruses during the induction of host symptoms development it is suggested that the increased energy demand during viral infection has a significant impact on energy metabolism, respiration, and photosynthetic rates.”.

Lines 395-397 – “These studies…symptoms development” – delete the sentence, as the idea is already given above.

Line 413-414 – reference Figure 3 at the end of line 415, for completion.

Line 422 – delete “in”.

Lines 436-438 – Maize is a C4 plant. So, high levels of malate and PPDK enzyme activity are expected under light. If SCMV is also affecting these levels a more detailed explanation is needed here, otherwise, what is exactly the effect being attributed to the viral infection?

Lines 511-512 – Please delete the sentence: “Looking ahead… virus-plants interactions”. It sounds like a buzzword and sentences like this just decrease the level of the whole manuscript. 

Line 540 – is this reference complete?

Reviewer 4 Report

This is a comprehensive Review and New Insights on the Complex Mechanisms Underlying Virus-Induced Symptoms in Plants. Understanding the mechanisms driving these symptoms is critical for devising effective disease management strategies. The review is well written and structured. However, the abstract is descriptive and did not clearly deliver the key massages of the study. Please at the end of each section add a summary sentence. I also suggested to extend the section of how the interactions between plant and virus are affected by environmental factors. Not only focus on temperature and light, but also extended it to other factors such as CO2, UV, ozone, drought…etc.

Round 2

Reviewer 1 Report

The authors took into consideration all of my comments. The paper looks good now. I accept it in its present form.

Minor English edit is needed. 

Reviewer 2 Report

Revised contents are satisfactory.